# Effects of dietary aromatase inhibitors on masculinization of rosy barb (*Pethia conchonius*): Evidence from growth, coloration and gonado-physiological changes

**Jham Lal, Pradyut Biswas\*, Soibam Khogen Singh ⓘ\*, Reshmi Debbarma, Suparna Deb, Nitesh Kumar Yadav ⓘ, Arun Bhai Patel**

Department of Aquaculture, College of Fisheries, Central Agricultural University, Lembucherra, Tripura, India

\* pradyutbiswas@gmail.com (PB); gengang@gmail.com (SKS)

**Data Availability Statement:** All relevant data are within the paper.

## Abstract

The objective of this study was to reveal the growth, colouration and gonado-physiological changes due to the exogenous aromatase inhibitor (AIs) in an ornamental fish. 17α-methyl-testosterone (MT) and letrozole (LET) were used as potential AIs. The AI were supplemented with a gel-based feed (LET: 50, 100, 150 and MT: 12.5, 25, 37.5 mg/kg feed) in Rosy barb, *Pethia conchonius* fry. The fishes were reared in a 45-L glass tank using AI treated gel-based feed for 3 months. Growth in AI-based diets was reduced but the reduction was minimal compared to the control. At 25 mg/kg feed of 17 MT, the highest male proportion (84.72% 6.05%) was recorded, which was significantly higher (P≤0.05) than other groups. L\*, a\*, and b\* values showed that 17α-MT-fed groups had brighter coloration (P≤0.05). Histological sections showed that LET-17α-MT suppressed ovarian development, causing atretic oocytes. Testicular development was unaffected. 25 mg/kg-treated feed increased SOD, CAT, GST, and GPX. The AI (MT) at 25 mg/kg gel-based feed could therefore be utilised for masculinization without impacting growth, colour, and antioxidant activity of rosy barb, which serves the entire male population in the ornamental fish sector.

## 1. Introduction

Ornamental fish are gaining in popularity for a number of reasons, including their high aesthetic value as well as their incredible commercial relevance in the export trade on a global scale. Attractive coloration is a major determinant in the market price of ornamental fish and is much dependent on the fish skin pigmentation [1]. In many commercial ornamental fishes, males possess brilliant colouration and are more attractive compared to the females [2]. These in turn exaggerate the price of the male specimens, and therefore, propagation of all-male populations could be quite profitable. Methods such as hand sexing, hybridization, and the generation of super males are among those available for generating the desired sex population. Hormonal sex reversal is a significant approach used widely in aquaculture sex manipulation. There are many published work of the successful use of steroid hormones to induce functional sex in aquarium fish [3].

**Funding:** The work received financial support from the Department of Biotechnology, Ministry of Science and Technology, New Delhi, India.

**Competing interests:** The authors have declared that no competing interests exist.

Many fish species have had great success with the direct feminization or masculinization that comes from using hormone sex control [4]. AI are commonly used to treat post-meno-pausal women with ovarian and breast cancer, and it is widely known that they can cause fish to develop more male characteristics by blocking estrogen's effects on ovarian differentiation [5, 6]. The cytochrome P450 aromatase enzyme plays a role in sex differentiation in fish by turning androgens into oestrogens. Because oestrogens actively contribute to normal female sexual differentiation, whereas natural male hormones do not, this is a crucial stage in the administration of AI, which causes the masculinization. Consequently, commercially available synthetic AI like anastrozole, letrozole, fadrozole, and exemestane are used to achieve a male-only population. Thus, AI can cause fish to become more masculine by blocking the effects of oestrogen on reproduction [6].

A recognized AI in rats and humans, letrozole (CGS 20267) (LET) has also been tested *in vitro* for action in rainbow trout ovarian microsomes [7, 8]. A dose-dependent decrease in aro-matase activity of up to 90% can be seen when LET was administered [9]. Letrozole (LET), a medication in the triazole group, is a well-known powerful AI that has indirect effects [10, 11]. By binding competitively to the haeme of cytochrome P450 subunits, LET inhibits oestrogen production in tissues [12]. Parallelly, the androgen 17-methyltestosterone (17-MT) occurs nat-urally in fish (MT). It is a synthetic androgen that promotes both sexual maturation in men and muscular growth in men [13, 14]. The main commercial application of monosex popula-tions is hormonal and selective breeding, which has received a lot of attention. Here, we see three distinct types of masculinization processes: i) androgen-based direct masculinization; (ii) estrogen-based indirect masculinization; and (iii) a combination of hormonal treatment and offspring assessment to establish distinct genotypes (YY "Supermales"). Despite the fact that each method has its own set of advantages and disadvantages, direct androgen treatment is typically simple [15].

There are a number of issues with efficacy and acceptability of hormone therapy when it is added directly to fish feeds. A special method for handling such a challenging situation is the use of gel-based meals in aquaculture. Gel feed is a three-dimensional network that is formed through chemical or physical cross linking. A gel can be thought of as a highly elastic, rubber-like solid. Gel feed that are nutrient-dense, hard, and flexible are made in animal-attractive shapes to ensure feed acceptance in a short time [16]. Gel feed may be used as a delivery sys-tem. Gel feed is improvements in animal feed composition, and in particular, to improvements in feed for aquatic animals. In short, the gel feed has the advantages of being easily digestible, reducing feed waste, and improving mineral absorption. It also improves the taste of feed for better consumption and helps fishes & shrimps grow faster and healthy.

The rosy barb, generally referred to as *Pethia conchonius* (Hamilton 1822), is a small, well-liked freshwater cyprinid that is indigenous to India and other regions of South and Southeast Asia. Due to their hardiness, *P. conchonius* is another species that aquarium hobbyists prefer to maintain at home aquaria as ornamental fish. The species has a higher global demand, with an export value of $6.99 - $9.99 per piece. It has a wide range of environmental sensitivities, the capacity to colonize disturbed environments, trophic opportunistic, as well as rapid develop-ment rates.

Thus, the present work highlights an attempt to use novel AI for achieving the target, which is especially important in light of the market opportunities and need for establishing an all-male production of ornamental fish of global significance. Accordingly, the objective of this research was to investigate whether LET and MT could successfully induce masculinization in *P. conchonius* fish. Further, the change in the sex ratio, survival rate, growth performance, gonadal histology and physiological alterations due to the hormone is examined. The study's

findings may shed light on how effectively the new hormonal therapy can boost the ornamental fish industry.

## 2. Materials and methods

### 2.1. Animal ethics

The experimental animal keeping and welfare strictly followed the guidelines laid by the Committee for the Purpose of Control and Supervision of Experiments on Animals (CPCSEA), Ministry of Environment & Forests (Animal Welfare Division), Government of India. Also, due approval of the Institutional Ethics Committee (IAEC) of the College of Fisheries, CAU, Tripura was taken vide approval number CAU-CF/48/IAEC/2018/09a.

### 2.2. Experimental site and fish-keeping

The study was conducted at wet laboratory of the College of Fisheries, Central Agricultural University, Lembucherra, Tripura, India during 2022. Early fry of rosy barb (Mean length/ weight = 2.86±0.05 cm/0.254±0.01 g) were randomly collected from the breeding tanks of the institute. Subsequently, they were transferred carefully to the experimental aquarium tanks which was previously filled with iron-free tap water and aerated over 3 days. Fishes were acclimated for 4–5 days in the experimental tanks and given planktonic live food collected from the ponds. Fishes were stocked at 10 fry per 40 L of usable water volume. Experiment was conducted in triplicates with 30 fish in each tank and fed with the experimental diet for 90 days. The feed was given to the stocked fishes at the rate of 7% of their body weight daily with equal ration (0900 hr/1600 hr). Left out feed and accumulated faecal matter was siphoned out daily morning in order to maintain good water quality. The water quality parameters maintained the optimum range during the experiment period. The water quality parameters were measured using the standard procedure outlined by APHA, (2005) [17].

### 2.3. Experimental design

The experimental design follows a completely randomized design and performed in triplicates (Table 1). Six experimental diets prepared using two AIs i.e., letrozole and 17 α–MT applied in three doses comprised the experimental treatments. The groups and accordingly designated as:

*Group 1* (T1): Feed supplemented with 50 mg/kg Letrozole
*Group 2* (T2): Feed supplemented with 100 mg/kg Letrozole
*Group 3* (T3): Feed supplemented with 150 mg/kg Letrozole
*Group 4* (T4): Feed supplemented with 12.5 mg/kg 17α-MT
*Group 5* (T5): Feed supplemented with 25 mg/kg 17α-MT

**Table 1. Experimental design and gel-based diets used.**

| Treatment | diets | Level of aromatase inhibitor in feed | | Replicates | |
|---|---|---|---|---|---|
| T1 | LET | Feed supplemented with 50 mg/kg Letrozole | R1 | R2 | R3 |
| T2 | LET | Feed supplemented with 100 mg/kg Letrozole | R1 | R2 | R3 |
| T3 | LET | Feed supplemented with 150 mg/kg Letrozole | R1 | R2 | R3 |
| T4 | MT | Feed supplemented with 12.5 mg/kg 17α-MT | R1 | R2 | R3 |
| T5 | MT | Feed supplemented with 25 mg/kg 17α-MT | R1 | R2 | R3 |
| T6 | MT | Feed supplemented with 37.5 mg/kg 17α-MT | R1 | R2 | R3 |
| T7 | None | Feed without aromatase inhibitor (control) | R1 | R2 | R3 |

*Group 6* (T6): Feed supplemented with 37.5 mg/kg 17α-MT

*Control* (T7): Feed without AI

## 2.4. Preparation of gel feed

Raw ingredients were selected according to requirement for the preparation of gel feed including fish muscle, corn flour, yeast, vitamin-mineral premix, lactogen, gelling agents and salt (Table 2). Required raw ingredients were measured for the preparation of gel feed. Fish muscle was mixed with salt, corn flour, yeast, lactogen, vitamin-mineral premix, and gelling agent (starch). Mixed wet ingredients were taken in an aluminum foil and sealed with sealing machine. Heating is important substitute for the preparation of gel feed were provided heat at water bath on 40°C for 2.5 hr. Gel feed were transferred/ kept in deep freezer for one night for proper setting of gel. For the current study, fishes were fed diets fortified with varying concentrations of AI. The concentration for MT treatment was also determined by our previous observations [18].

## 2.5. Growth performance

Fish were chosen randomly from every tank and measured for their weight and length. Every treatment group's average body weight, percentage weight gain, as well as specific growth rate, were measured following the formulae [19].

**Average body weight gain (g)**

$$\text{Body weight gain} = \textit{Average final weight} - \textit{Average initial weight}$$

**Average body Length gain (cm)**

$$\text{Body weight gain} = \text{Average final length} - \text{Average initial length}$$

**Percentage weight gain (%)**

$$\text{Percentage weight gain} = \frac{\text{Average final weight} - \text{Average initial weight}}{\text{Average initial weight}} \times 100$$

**Specific growth rate (%)**

$$\text{Specific growth rate/day (\%)} = \frac{(\text{In final weight} - \text{In initial weight})}{\text{Number of days}} \times 100$$

**Table 2. Ingredient composition of experimental gel-based feed for *P. conchonius*.**

| S.N. | Ingredient composition | (g) |
|---|---|---|
| 1 | Fish muscle (g) | 80 |
| 2 | Corn flour (g) | 30 |
| 3 | Starch (g) | 5 |
| 4 | Lactogen (g) | 5 |
| 5 | Yeast (g) | 2 |
| 6 | Vitamin-mineral premix (g) | 2 |
| 7 | Vitamin C (g) | 2 |
| 8 | Salt (g) | 2.5 |

## 2.6. Fish sampling procedure

After the trial, fishes were harvested from each individual tank using a hand net. They were given mild anaesthesia (clove oil at 50 μg. L$^{-1}$) immediately. Blood was drawn carefully from the caudal peduncle region with a 1 mL hypodermal syringe (24-gauge needle) and was collected in the Eppendorf tubes without EDTA, and allowed to clot for 2 h, which was then centrifuged (3000 × g for 15 min).

## 2.7. Skin colour intensity

Using a colour analyser (HunterLabTM, Hunter Associates Laboratory, Inc. USA), the reflected light from the dorsal skin of rosy barb fish was measured and compared through stranded calibration [20]. Fish were anaesthetized with diluted clove oil prior to the measurement of colour intensity while they were kept in the sample holding dish, and they were then reintroduced after the values were obtained. According to Yesilayer and Erdem (2011) [21], the colour parameters lightness (L), greenness-redness chromaticity (a*), and yellowness-blueness chromaticity (b*) were evaluated in accordance with the International Commission on Illumination's standards. The value of "L" is a number between 0 and 100. Absolute brightness is represented by a value of "100" for "L," whilst absolute darkness is represented by a values of "0." a* and b* have values that range from + to -. While negative values of "a*" and "b*" define green and blue colours, respectively, a substantially greater positive "a*" and "b*" value represents a superior level of redness and yellowness-orange colour. L stands for lightness and is determined using the 'Y' tristimulus utilising Priest's approximation of the Munsell value as follows:

$$L = 100 \ (Y/Yn)$$

The "Yn" value denotes the Y tristimulus of a recognised white item. Opposing colour axes are denoted by the symbols "a*" and "b*." 'a*' stands for 'Redness,' which is (+, positive) compared to 'Greenness,' which is (, negative), and is computed as:

$$a = Ka(X/Xn - Y/Yn)/\sqrt{(Y/Yn)}$$

Where "Ka" stands for an illuminant-dependent coefficient and "Xn" stands for the white object's "X" tristimulus,

The opponent colour axis (b*), which is positive (+) for yellow colours and negative () for blue colours, is determined as follows:

$$b = Kb(Y/Yn - Z/Zn)/\sqrt{(Y/Yn)}$$

Where "Zn" stands for the Z tristimulus of the specified white object and "Kb" is a variable that relies on the illumination source.

Whiteness was determined using the techniques mentioned [22]:

$$\text{Whiteness} = 100 - [(100 - L*)^2 + a*^2 + b*^2]^{1/2}$$

## 2.8. Whole body carotenoid content

Carotenoid extraction was carried out using the Torrissen and Naevdal methodology (1988) [23]. Briefly, acetone (10 mL), anhydrous Na$_2$SO4, and fish tissue (0.1–1 g) were combined together (2 g). After centrifuging the homogenate at 3500 g for 5 min, the supernatant was collected and stored at 4˚C. Using a spectrophotometer, the absorbance of cold supernatant was determined at 476 nm (Thermo-Scientific, India). By multiplying the absorbance at 460 nm by

the dilution factor, the total carotenoid concentrations (g/g tissue) were determined (0.25 x weight of sample in g).

## 2.9. Antioxidant enzyme activity

The levels of various enzymatic antioxidants, including superoxide dismutase (SOD), catalase (CAT), glutathione reductase (GRD), glutathione peroxidase (GPX), and glutathione S-transferase (GST), were assessed in the serum of fish from various treatment groups (n = 30). Blood samples were taken from the fish's caudal vein using a sterilized insulin syringe devoid of anticoagulation, as well as then they were centrifuged (3,000 rpm, 10 min) to separate the serum [24].

SOD activity was estimated using a spectrophotometric method [25] based on the evaluation of $O_2$ facilitated NBT decrease with just an anaerobic combination of NADH and phenazine methosulphate (PMS). To determine CAT activity, absorbance was determined at 240 nm for up to 90 seconds at frequent 15-second intervals [26]. By treating the serum with sodium azide, a very well inhibitor of CAT activity, the enzymatic assay was validated [27]. To determine the amount of GPX activity, the absorbance was determined at 492 nm against with a blank (100 l additional O-phenylenediamine solution instead of sample) [28]. Utilizing glutathione (GSH, 2.4 mM/l) and 1-chloro-2, 4-dinitrobenzene (CDNB, 1 mM/l) as substrates, GST activity was measured spectrophotometrically [29]. By combining 150 ml of Ellman's reagent with 50 ml of serum, the GSH level was calculated. After 25 minutes, absorbance at 414 nm was measured to determine the GSH level [30].

## 2.10. Sex ratio and histology

Visual examination of the external morphological characteristics served as the primary method for sexing the fish from various treatment groups [31]. In order to confirm sex, fish were slaughtered and dissected. Male fish's testicles and female fish's ovaries were visually examined under microscpe at 10 X and 40 X magnifications. The histological examination was conducted on five fish that had received AI treatment and five control fish at every sampling time point. After dissection, a sex examination of the gonadal tissue was carried out. The gonads were preserved in Bouin's solution for histological inspection after the sex of *P. conchonius* had been tentatively identified. The tissue structure was examined underneath a microscopic examination and captured photographs through the camera after the gonadal (ovary and testis) wereas embedded in paraffin, sectioned (5 mm thick), and stained with hematoxylin-eosin (HE). All samples were categorized into two phases of male and female gonad development based on the documented developmental stages of the samples' gonads and existing histology examinations of the gonads of *P. conchonius.*

## 2.11. Gonadosomatic index

The gonads of *P. conchonius* both male and female were taken out, weighed to the nearest 0.02–0.04 g in order to calculate the GSI, and examined under a compound light microscope to look for signs of ovarian development. The remainder of the fish received the control food and was left in their individual tanks [32]. The gonadosomatic index (GSI) of matured fish was determined using the formula:

$$\text{GSI (Gonado Somatic Index) (\%)} = \frac{(Gonad\ weight\ (g)}{Body\ weight\ (g)}\ X\ 100$$

## 2.12. Statistical Analysis

The experimental data was sorted by Excel and subjected to a One-way ANOVA utilizing software (IBM SPSS Statistics version 21.0 for Windows) to ascertain whether there were any significantly different between the different treatment groups. 95% possible outcome reconciliation was performed (P≤0.05). The data were summarized as mean ± standard deviation (SD). A posthoc, Duncan's options, descriptive comparisons test was performed after the one-way ANOVA analysis when substantial differences were discovered.

# 3. Results

## 3.1. Growth performance

After 3 months of culture, all treatment and control groups had similar survival rate as presented in Fig 1. Survival ranged from 83.33 in control to 93.33% in treated groups throughout the experiment. The lowest mortality rate in the MT-treated group (25mg/kg) was 6.66%. Control fish had the highest body weight gain (2.70 ± 0.01) and % weight gain (1054.78±8.96) (Table 3). Control group fish had a significantly (P≤0.05) higher specific growth rate than other treatment groups (1.18±0.003). The length gain was not significantly different (P≤0.05) between the control (2.63±0.05) and experimental (2.60±0.10 to 2.90±0.26 cm) groups after 90 days.

## 3.2. Skin colour intensity and whole body carotenoid content

The colour intensity of the skin and the carotenoid content of the entire body differed significantly between treatment groups, as shown in Table 4. The lightness was significantly higher (P≤0.05) in the 100 mg/kg LET gel-based feed groups and significantly lower in the 50 mg/kg LET gel-based feed groups. Skin colour intensity in terms of redness and yellowness was found to be highest in fish fed 25 mg/kg MT in the diet, with 'a*', index of redness (37.47±0.93a) and 'b*', index of yellowness (34.7±11.58a) that were significantly different from all other treatment groups. Whole-body carotenoid content was highest in the 37.5 mg/kg MT (9.21±6.76a) and lowest in the (2.63±0.17b) control groups. Furthermore, the highest whiteness (41.26±4.53a)

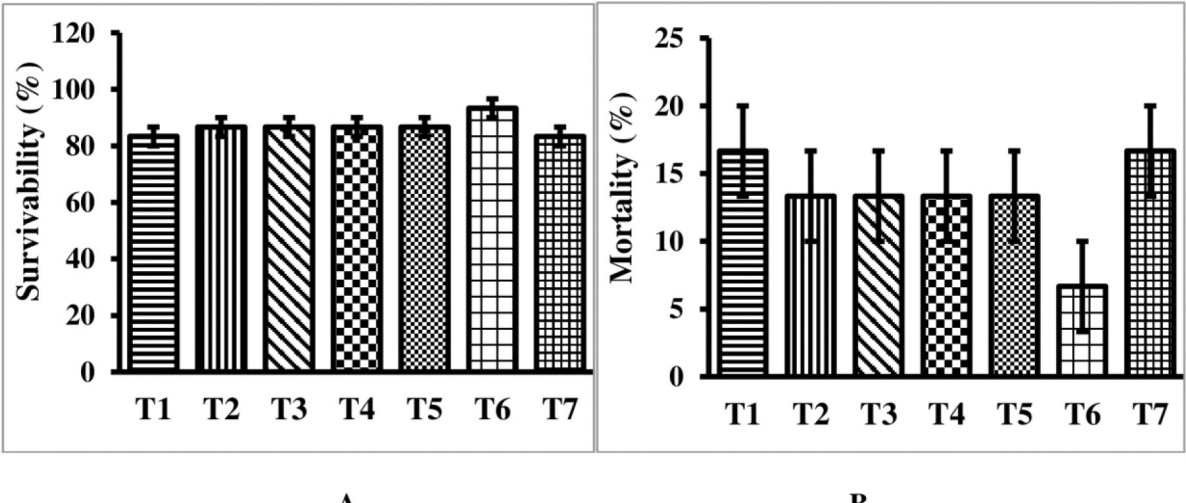

**Fig 1. Survivability and mortality of rosy barb, *P. conchonius* at end of the musculinization experiment. A.** Survivability; **B.** Mortality. *Data are represented as mean ± SE. Different superscripts indicate statistically significant difference (P≤0.05) among the experimental groups.

**Table 3. Effect of Letrozole and 17α-methyltestosterone on growth performance and GSI of rosy barb, *P. conchonius*.**

| Treatments | Body weight gain | Body length gain | % weight gain | Specific growth rate | Gonadosomatic Index (Male) | Gonadosomatic Index (Female) |
|---|---|---|---|---|---|---|
| T1 | 2.59±0.06b | 2.77±0.11 | 1004.44±5.92b | 1.15±0.002b | 1.47±0.26 | 9.66±0.03 |
| T2 | 2.33±0.02c | 2.67±0.20 | 953.12±3.88c | 1.13±0.001c | 1.50±0.08 | 9.63±0.02 |
| T3 | 2.32±0.04c | 2.67±0.15 | 919.77±3.83d | 1.12±0.001d | 1.59±0.15 | 9.61±0.01 |
| T4 | 2.27±0.10c | 2.67±0.11 | 892.17±3.74e | 1.10±0.001e | 1.56±0.02 | 9.64±0.002 |
| T5 | 2.14±0.02d | 2.63±0.05 | 884.82±2.41e | 1.10±0.001e | 1.66±0.08 | 9.62±0.03 |
| T6 | 2.06±0.03d | 2.60±0.10 | 840.26±6.66f | 1.08±0.003f | 1.70±0.12 | 9.59±0.09 |
| Control | 2.70±0.01a | 2.90±0.26 | 1054.78±8.96a | 1.18±0.003a | 1.44±0.08 | 9.89±0.63 |
| *P*-value | | | | | | |
| AI | < 0.001 | 0.388401 | < 0.001 | < 0.001 | 0.079283 | 0.898913 |
| Conc. | < 0.001 | 0.099778 | < 0.001 | < 0.001 | 0.336924 | 0.938719 |
| AI×Conc[1] | < 0.001 | 0.771600 | < 0.001 | < 0.001 | 0.884706 | 0.999403 |

Values are means ± SD, n = 3 per treatment group. [a-f] Means in a column without a common superscript letter differ (P≤0.05) as analyzed by two-way ANOVA and the DUNCAN test.

[1]AI × Conc = Aromatase inhibitor × Concentration interaction effect.

was observed in the 150 mg/kg LET diets; there was a significant (P≤0.05) difference between the control and other treatment groups.

### 3.3. Antioxidant activity

SOD (11.29±0.005a U/mg protein/min), GST (6.78±0.01a U/mg protein/min), GPX (3.41 ±0.005a U/mg protein/min), and the lowest value of GST (1.18±0.02a nmol/mg protein) were found in 37.5mg/kg MT and were significantly (P≤0.05) different from all other groups (Table 5). The 25mg/kg MT had the highest CAT (1.18±0.02a nmol/mg protein); there was a significantly (P≥0.05) difference between all groups (Table 5).

### 3.4. Sex ratio and histology

At the end of the three-month experiment, there was a significant (P≥0.05) difference in the male/female sex ratio (%) among all treatment groups (Fig 2). Among the MT-treated groups,

**Table 4. Colour analysis of rosy barb, *P. conchonius* at the end of musculinization period.**

| Treatments | L* (Lightness) | a* (Redness/greenness) | b* (Yellowness/blueness) | Whiteness | Carotenoids |
|---|---|---|---|---|---|
| T1 | 47.02±1.35b | 27.86±4.97c | 29.37±5.74abc | 33.19±5.78bc | 2.98±0.06b |
| T2 | 57.90±2.28a | 32.25±0.06bc | 29.19±0.19bc | 39.45±1.71ab | 3.50±0.81b |
| T3 | 56.68±5.83a | 28.34±2.31c | 27.44±2.55cd | 41.26±4.53a | 3.89±1.23b |
| T4 | 53.16±4.27ab | 30.91±0.93bc | 31.69±2.55abc | 35.52±4.33abc | 3.98±1.07b |
| T5 | 50.24±1.56b | 37.47±0.93a | 33.43±1.66ab | 29.29±1.75c | 5.01±0.74ab |
| T6 | 51.68±3.58ab | 34.83±2.19ab | 34.71±1.58a | 30.99±2.44c | 9.21±6.76a |
| Control | 50.25±1.18b | 28.51±1.31c | 23.59±2.43d | 37.96±1.18ab | 2.63±0.17b |
| *P*-value | | | | | |
| AI | 0.184416 | 0.000542 | 0.004173 | 0.002638 | 0.056427 |
| Conc. | 0.082675 | 0.004197 | 0.889647 | 0.614290 | 0.153844 |
| AI×Conc[1] | 0.006426 | 0.455288 | 0.347771 | 0.010701 | 0.336807 |

Values are means ± SD, n = 3 per treatment group. [a-d]Means in a column without a common superscript letter differ (P≤0.05) as analyzed by two-way ANOVA and the DUNCAN test.

[1]AI × Conc = AI × Concentration interaction effect.

**Table 5. Effect of Letrozole and 17α-methyltestosterone on antioxidant activity of rosy barb, *P. conchonius*.**

| Treatments | Glutathione peroxidase (GPX) | Glutathione S-transferase (GST) | Catalase (CAT) | Superoxide dismutase (SOD) |
|---|---|---|---|---|
| T1 | 11.24±0.01cd | 6.73±0.005d | 1.15±0.01bc | 3.37±0.01b |
| T2 | 11.26±0.01bcd | 6.74±0.01d | 1.16±0.01ab | 3.38±0.005b |
| T3 | 11.27±0.02ab | 6.76±0.005c | 1.17±0.01a | 3.39±0.01ab |
| T4 | 11.25±0.01bcd | 6.76±0.005bc | 1.16±0.01ab | 3.39±0.01ab |
| T5 | 11.27±0.01abc | 6.77±0.01ab | 1.17±0.01a | 3.40±0.02a |
| T6 | 11.29±0.005a | 6.78±0.01a | 1.18±0.02a | 3.41±0.005a |
| Control | 11.24±0.01d | 6.73±0.005d | 1.14±0.02c | 3.35±0.01c |
| *P*-value | | | | |
| AI | 0.121776 | 0.000003 | 0.097664 | 0.001824 |
| Conc. | 0.006393 | 0.002958 | 0.036360 | 0.076813 |
| AI×Conc[1] | 0.973284 | 0.571038 | 0.890738 | 0.905478 |

Values are means ± SD, n = 3 per treatment group. [a-d]Means in a column without a common superscript letter differ (P≤0.05) as analyzed by two-way ANOVA and the DUNCAN test.

[1]AI × Conc = Aromatase inhibitor × Concentration interaction effect.

the group treated with MT at 25mg/kg had a significantly higher (P≥0.05) percentage of males (82.22%). The control group, on the other hand, had the lowest percentage of males (43.98%), while females accounted for 56.01% (Fig 1). The percentage of male and female *P. conchonius* in the MT treated groups (with the exception of 25mg/kg) was significantly (P≤0.05) different from the control groups.

In the present study, fifty-four females and one hundred twenty-eight males were found among the one hundred eighty-two specimens. The developing ovaries in females had thicker walls, larger oocytes, more regular egg sizes, an orange colour, and 0.24 g of gonad weight. The majority of the ovaries are in late stage IV to late stage V, according to the gonad tissue sections. The majority of the yolk has accumulated as the nucleus is compressed (Fig 3). The original gonad of a male *P. conchonius* weighs 0.04 g, is off-white with a lot of semen, and is thin (Fig 4). Generally speaking, stem cells, secondary spermatocytes, and sperm cells make up

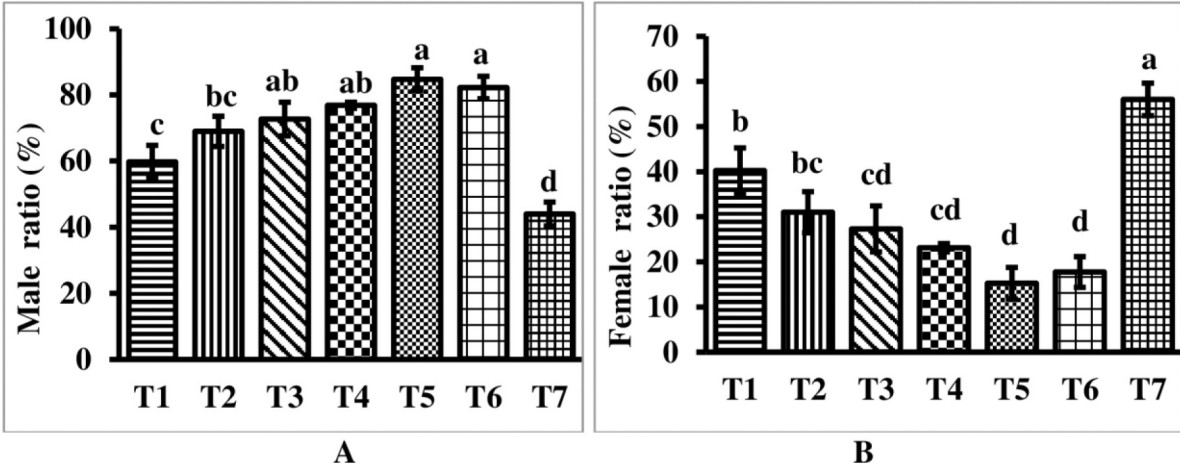

**Fig 2. Male and female ratio of *P. conchonius* at end of the musculinization experimental trial. A.** Male; **B.** Female. *Data are represented as mean ± SE. Different superscripts indicate statistically significant difference (P≤0.05) among the experimental groups.

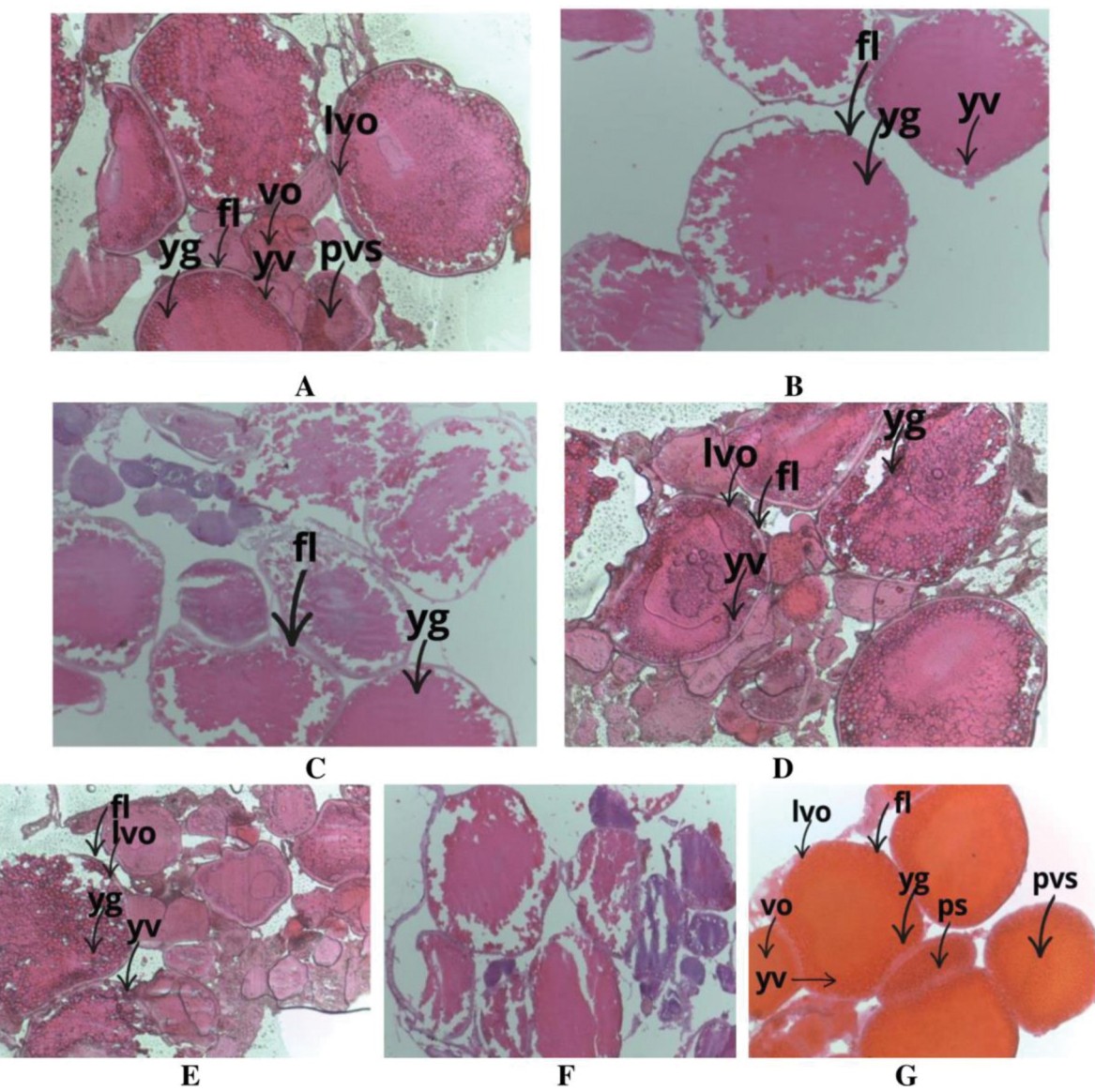

**Fig 3. Histological study of *P. conchonius* ovary photomicrograph (10X) after 90 days of masculinization period. A:** 50 mg/kg LET; yolk vesicle (yv), yolk granules (yg), vitellogenic oocyte (vo), follicular layer (fl), late vitellogenic oocyte (lvo), pre-vitellogenic stage (pvs); **B:** 100 mg/kg LET; yolk vesicle (yv), yolk granules (yg), follicular layer (fl); **C:** 150 mg/kg LET; follicular layer (fl), yolk granules (yg); **D.** 12.5 mg/kg MT; yolk vesicle (yv), yolk granules (yg), follicular layer (fl), late vitellogenic oocyte (lvo); **E:** 25 mg/kg MT; yolk vesicle (yv), yolk granules (yg), follicular layer (fl), late vitellogenic oocyte (lvo), **F:** 37.5 mg/kg MT treated showing degenrated atretic oocytes (ao); **G:** perinucleolar stage (ps), yolk vesicle (yv), yolk granules (yg), vitellogenic oocyte (vo), follicular layer (fl), late vitellogenic oocyte (lvo), previtellogenic stage (pvs).

gonad seminal vesicles under tissue sections. When there are enough mature sperm in the lobular cavity, the gonads begin to mature.

### 3.5. GSI

Table 3 shows the results of the oral and immersion treatment of Letrozole and MT on male and female GSI. There was no significant ($P \leq 0.05$) difference in female GSI between the treated and control groups in the oral group. The male GSI of the treated groups was

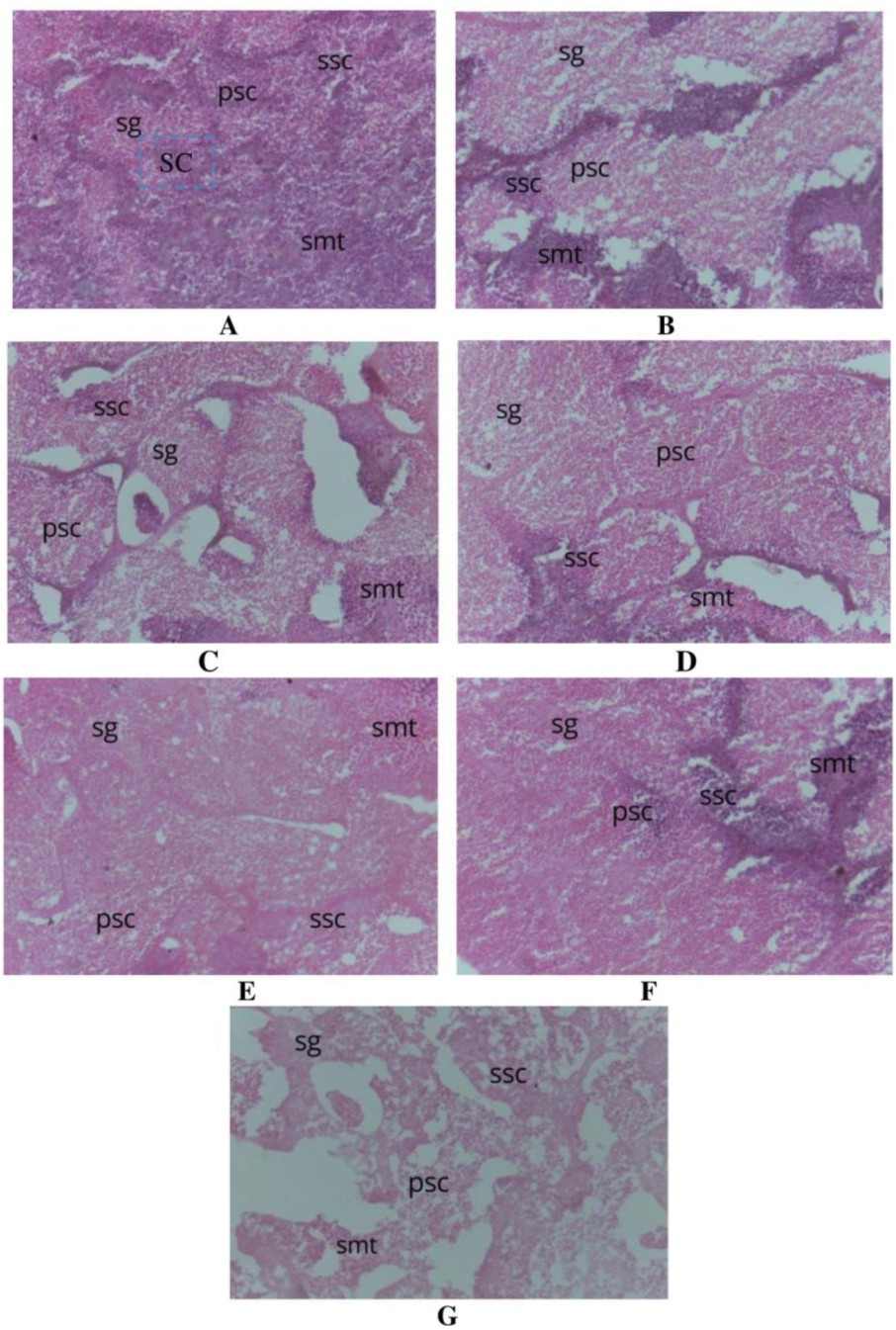

**Fig 4. Histological study of *P. conchonius* testis photomicrograph (40X) after 90 days of musculinization period.**
**A:** 50 mg/kg LET spermatogonia (sg), primary spermatocytes (psc), secondary spermatocytes (ssc), (smt) and spermatozoa; **B:** 100 mg/kg LET (sg), (psc), (ssc), spermatids (smt) and spermatozoa; **C:** 150 mg/kg LET (sg), (psc), (ssc), (smt) and spermatozoa; **D.** 12.5 mg/kg MT (sg), (psc), (ssc), (smt) and spermatozoa; **E:** 25 mg/kg MT (sg), (psc), (ssc), (smt) and spermatozoa; **F:** 37.5 mg/kg MT (sg), (psc), (ssc), (smt) and spermatozoa; **G:** Control (sg), (psc), (ssc), (smt) and spermatozoa.

significantly (P≤0.05) lower than the control group, indicating that ovarian development was suppressed. Females had the highest GSI in the oral treatment, at 37.5 mg/kg MT, compared to 21.5 mg/kg MT in the control groups. The female fish GSI was higher in the control groups

was observed compared to the treated groups, suggesting that ovarian development was slightly suppressed in the treatment groups, with a significantly lower value at 25 mg/kg. When compared to the control, both male and female GSI were significantly altered (P≤0.05).

## 4. Discussion

### 4.1. Growth performance

There have been reports of 17-MT having anabolic effects on ornamental fish growth; however, these effects vary depending on the time, hormone dosage, and length of therapy. In the present study, we noticed a growth reduction in AI treated groups compared to control. We observed that fish growth was apparently affected by the 17α-MT dosages utilised, whereas the LET-treated group showed the maximum level of growth inhibition. The serum 17-MT levels obtained may have been just above the upper limit for growth promotion. In contrast, Othman et al. (2022) [33] reported the highest growth performance parameters such as the highest body weight gain, and SGR after the feeding of 50 mg/kg MT-treated feed at 38 and 46 days. Bautista et al. (2022) [34] also recently examined the weight gain and feed acceptance by the fish were higher in male catfish fed diets containing 60 and 90 mg•kg-17- MT. Similar findings have been made in other fish species such as Nile Tilapia, *Oreochromis niloticus* [35], and Emperor red cichlid, *Aulonocara nyassae* [36].

### 4.2. Colour analysis

The quality and attractiveness of ornamental fish are defined by their skin colour, which can influence demand and value during global trade [37]. A distinct coloration (red/green and yellow/blue colour development) was observed in diets containing 17a-MT. Coloration improved and increased with increasing hormone concentrations until it peaked in the 17-MT hormone group. The control, 17-MT, and LET AI-treated groups were found to be significantly different. The substance caused more colour changes in the fish than in the control fish. The effects of hormones on the fish's neuro-peptide system may be responsible for the colour change. The hormone (MT) stimulates the pituitary gland to produce more melanophore dispersing hormone (MDH), which increases the concentration in the bloodstream and causes colour variations depending on hormone concentrations [38]. Larsson et al. (2002) [39] found more reddish tail coloration in guppies on the eighth day of experimental studies with 17a-MT treatments, with significant differences from the control group on the 18th day. Color formation in guppies began after the third week of 17a-MT hormone experimental studies, according to Keskin (2005) [40]. This study concluded that such hormone therapies could be used to produce male fish at any time and in as large a quantity as possible, with dazzling colours and higher economic values. When colour analyses were first recorded in this study, the fish were still in their early stages of development, and hue (H˚ab) readings suggested shades of grey-brown. There were visible colour differences between groups based on the samples analyzed on day 30. At the end of the trial, there were differences in desired colour between the 17-MT-supplemented fish groups and the control groups.

### 4.3. Anti-oxidant activity

Antioxidant enzymes, such as SOD and CAT, are believed to be the first line of defence against oxidative disequilibrium and tissue damage [41]. SOD and CAT eliminate excess hydrogen peroxide from the tissue and scavenge superoxide molecules. We observed a difference in the SOD and CAT values between the two experimental groups, with the AI-treated groups exhibiting higher values. In general, the production of the hydrogen peroxide radical induces

oxidative stress in animals, necessitating the activity of SOD and CAT. However, the performance of these enzymes may be low due to depletion of their action under conditions of high oxidative stress. When these radicals are present in low concentrations, the production of these enzymes is stimulated, and we can see them in greater quantities. Zheng et al. (2016) [42] found that 0.5 mg/L MT significantly reduced antioxidant activity SOD, CAT, and GST, as well as transcriptional changes (SOD and CAT) in the liver juvenile of tilapia, *O. niloticus*. Other antioxidant enzymes such as T-GSH, GSH, and MDA were also significantly reduced by 0.5 mg/L MT treatment on the 7th and 14th day. While the results obtained here are intriguing, due to a lack of prior work, comparisons cannot be drawn at this time, necessitating further research on how AI diets affected the antioxidant system.

## 4.4. Sex ratio and histology

The distinction between gonadal sex and sex change in several fish species was previously demonstrated to depend critically on ovarian aromatase and estrogens [43]. This research's main objective was to examine how the given AIs affected sex changes and found that the highest concentration of 17-MT (25 mg/kg) was produced in the largest percentage of males. There was no evidence of intersex in this investigation. In fish groups treated with 25mg/kg 17-MT, the masculinization rate was determined to be 84.72%. In accordance with previous research by Karaket et al. (2023) [44], 17-MT at 1000 to 2000 g/L produced significantly more male offspring than the expected 1:1 sex ratio. It was observed that a few component combinations had an impact on how masculine red tilapia appeared. Significant changes in the male-to-female ratio were made by the dose-period and dose-frequency treatments. In another study, 50 mg/kg and 100 mg/kg were both examined. The 17-MT treated feed deviated from the anticipated typical male: a female population at 38 and 46 dph (1:1). The 50 mg/kg 17-MT dosage feed groups fed with 17-MT at 67 dph produced fish that were 37% and 25% intersex, respectively [33]. The fish's sex ratio did not change across treatment groups; however Chitosan nanoparticles increased the effectiveness of LET. The use of chitosan nanoparticles for LET delivery, according to the authors, could decrease estradiol synthesis, enhance testosterone levels, and influence the sex ratio in rainbow trout [45]. According to Karaket et al. (2023) [44], 1000 g/L of 17 MT administered for 6 hr once a week produced the highest male proportion (95.33 ±0.58%), which was significantly higher than the results of the other groups. The male ratio was significantly affected by the dose-period and dose-frequency treatments. The indirect administration of 500 mg/kg 17-methyltestosterone (17α-MT) to XX mandarin fish with fully developed ovaries (60 dpf, days after fertilization) resulted in secondary sex reversal. [46]. The Zebrafish International Resource Center frequently supplemented larval nutrition with 17-methyltestosterone to push gonadal sex differentiation towards masculinization, yielding up to 80% males on average across stocks [47]. e Silva e al. (2022) [48] reported the 94% masculinization in tilapia after the fed of 17α-MT at 60 mg kg-1 five times daily on the BFT systems. When given 50 mg/kg of 17α-MT, an exogenous hormone, the juvenile yellow perch, *Perca flavencens*, generated up to 100% male offspring in contrast to the other treatment groups [33].

The previous studies of the gonad histological differentiation [49] are frequently the most efficient period for the hormone to produce sex-reversed fish, including bluegill sunfish [50, 51] and yellow catfish [52]. In the current study, primordial germ cells, spermatogonia (sg), primary spermatocytes (psc), secondary spermatocytes (ssc), spermatids (smt), and spermatozoa were found to proliferate in the testis of *P. conchonius*. In the case of the ovaries, 17-MT inhibited ovarian development, resulting in atretic oocytes. Similarly, Katare et al. (2021) [53] revealed the AI masculinization increased the proliferation of spermatogonia (sg), primary

spermatocytes (psc), secondary spermatocytes (ssc), spermatids (smt), and spermatozoa in the testis of *Trichogaster lalius*.

In the case of female musculinization, AI has stopped the secreation of the aromatase enzyme, so, therefore released the testis development hormones to help the development of testis in the fish. According to the number of studies [54], an animal must be fed LET at least twice a day in order to be successfully "masculinized". The possibility existed that all of the MT in the feed would be absorbed, leading to an excess of hormone being administered, based on the original hypothesis that the fish would receive more gel-based diet constantly. Because MT has a short half-life, it's possible to increase the quantity of feed consumed by the fish undergoing oral masculinization therapy to increase the concentration of hormones in the blood. As a result, there may be a greater masculinization of the female fish, which could reduce the number of female fish [55]. These findings suggest that estrogen regulates gonadal differentiation by directly influencing the development of testis differentiation factors in several fish species. A crucial step in the formation of the testis is the proliferation of spermatogonia. Histological examinations in this work revealed that cells first began to proliferate on the dorsal outermost surface of the ovaries when AI made *P. conchonius* more masculine.

### 4.5. Gonado-somatic index

The GSI is a crucial component for assessing an animal's ability to reproduce, and it is helpful for both scientific studies of reproductive biology and brood management in the aquaculture sector. The fish's gonads (ovary and testis) are mature if the GSI scores are elevated. The AIs groups in the present study had greater GSI values than the other groups. The GSI value is influenced by both the AI concentration and feeding frequency. According to the findings of some prior researchers, including those by Katare et al. (2021) [53], revealed the concentration of anastrozole was increased, the testicles' growth was enhanced, but the female GSI was higher in the control group due to the atretic effects in the female fish. All AI-treated groups demonstrated GSI suppression as anastrozole dosages increased.

In a previous study, 17α-MT-treated *Trichogaster lalius* females showed evidence of GSI reduction [56]. In the 3mg 17α-MT/kg treated groups, the GSI index of rainbow trout, Oncorhynchus mykiss, ranged from 0.04 to 0.07 for females and 0.07 to 0.07 for males, respectively, according to Rastiannasab and Kazemi (2022) [57]. Male fish from the 17α-MT treated group with 100% sex reversed were given a GSI index of 0.06; nevertheless, there was no significant variance between both the treatment groups' male and female populations. The female fish 50 mg/kg 17α-MT treated feed was substantially lower GSI at 38 and 46 dph [33] compared to the different treatment groups. Male catfish fed the MT diet had greater and noticeably longer testes than male catfish fed the control diet at the end of the feeding trial. The GSI of these male catfish was likewise noticeably higher than that of the catfish in the control group [34]. In the anastrozole immersion and oral therapy groups, there was little inhibition of ovarian development. The gonadosomatic index was substantially higher in the protandrous fish, black porgy, and fadrozole group [58]. Because it is used to identify hydrated ovaries in females and testicles in males, the GSI contributes to distinguishing the reproductive season from weight increase [59].

## 5. Conclusion

High rates of masculinization were observed in *P. conchonius* at 25mg/kg 17 MT via oral administration of gel-based feed for up to 90 days, as shown in the current study. The simplicity, cost savings, and security of this protocol are more advantageous to operators. Hence, the masculinization processes used in the commercial male production of rosy barb can be used

directly to this gel-embedded oral delivery approach. Letrozol was used to musculinize the sex in *P. conchonius*, and all available evidence indicates that this may be the first report of its kind examining both sex change and physiological disturbances. There is a need for additional research to determine the most effective way of extending the duration of the oral treatment while simultaneously increasing the dosage in order to achieve complete masculinization.

## Acknowledgments

The authors would like to express their profound gratitude to the Vice Chancellor of Central Agricultural University, Imphal, and the Dean of the College of Fisheries at Central Agricultural University, Tripura, for providing the necessary facilities for the study. The corresponding author, Dr. Soibam Khogen Singh would like to acknowledge the assistance received from Institutional Development Plan (IDP)-NAHEP of the Central Agricultural University, Imphal, India for undergoing faculty foreign training at James Cook University, Singapore.

## Author Contributions

**Conceptualization:** Jham Lal, Pradyut Biswas, Soibam Khogen Singh.

**Data curation:** Jham Lal, Pradyut Biswas, Soibam Khogen Singh.

**Formal analysis:** Jham Lal, Pradyut Biswas, Soibam Khogen Singh.

**Investigation:** Pradyut Biswas, Soibam Khogen Singh, Arun Bhai Patel.

**Methodology:** Jham Lal, Pradyut Biswas, Soibam Khogen Singh.

**Supervision:** Pradyut Biswas, Soibam Khogen Singh, Arun Bhai Patel.

**Validation:** Pradyut Biswas, Soibam Khogen Singh, Arun Bhai Patel.

**Visualization:** Pradyut Biswas, Soibam Khogen Singh, Arun Bhai Patel.

**Writing – original draft:** Jham Lal, Reshmi Debbarma, Suparna Deb, Nitesh Kumar Yadav.

**Writing – review & editing:** Pradyut Biswas, Soibam Khogen Singh.

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
