## [Decision Letter · Decision Letter 0]

29 Mar 2023

PONE-D-23-08082Effects of dietary aromatase inhibitors on masculinization of Rosy Barb, Pethia conchonius: evidences from the growth, coloration, gonado-physiological changesPLOS ONE

Dear Dr. Singh,

Thank you for submitting your manuscript to PLOS ONE. After careful consideration, we feel that it has merit but does not fully meet PLOS ONE’s publication criteria as it currently stands. Therefore, we invite you to submit a revised version of the manuscript that addresses the points raised during the review process.

We look forward to receiving your revised manuscript.

Kind regards,

Dharmendra Kumar Meena

Academic Editor

PLOS ONE

Journal Requirements:

Additional Editor Comments (if provided):

The authors have attempted and rightly put the things in right way. However I recommend to revise the discussion and conclusion section and also whole MS for language and phrases mistakes and other shortfalls.

Reviewers' comments:

Reviewer's Responses to Questions

**Comments to the Author**

1. Is the manuscript technically sound, and do the data support the conclusions?

Reviewer #1: Yes

Reviewer #2: Yes

2. Has the statistical analysis been performed appropriately and rigorously? 

Reviewer #1: Yes

Reviewer #2: Yes

3. Have the authors made all data underlying the findings in their manuscript fully available?

Reviewer #1: No

Reviewer #2: Yes

4. Is the manuscript presented in an intelligible fashion and written in standard English?

Reviewer #1: Yes

Reviewer #2: No

5. Review Comments to the Author

Reviewer #1: This study shows the role of two aromatase inhibitors,17-methyltestosterone (MT) and letrozole, in enhancing reproduction, growth, coloration and physiological profile changes. I tried to connect the work to previous reports and noticed that many work on MT have been reported, whereas the second AI, letrozole, is a new entrant in fish studies. The work's novelty, in which gel-based diets are introduced as a delivery vehicle, is applauded. Earlier studies on

physiological consequences and coloration were also not well defined, as attempted in this work.

I reviewed the work presented and believe that it can be published after careful refinement of the

technical parameters and writing as detailed point by point as possible.

Title: "Effects of dietary aromatase inhibitors on Rosy Barb, Pethia conchonius, masculinization:

evidence from growth, coloration, and gonado-physiological changes."

Line 34, "for 90 days," can be removed.

Lines 39-42: Rewrite the sentence.

Introduction

Aromatase inhibitors' second mention in this section and elsewhere may be abbreviated.

Lines 83-93 may include recent reports on the use of gels in fish feed.

Materials and procedures:

Known units can be written in abbreviated form. Hours, for example, can be written as hr.

Line 207: Include a reference.

Section 2.9: Magnification is required.

Statistical evaluation: Were the results normalized? Mention the methods that were used.

Results:

P values can be written in a consistent manner. It has been noticed that it is written as "P" and

"p" in many places.

Discussion

The discussion is quite detailed. To explain the current finding, the authors must concentrate

solely on related works. To strengthen the findings, delete any redundant sentences in this

section. Rather than indicating and comparing with previous studies, present the most likely

reasons why this occurred. The use of words like "mild" is highly unscientific. Change all such

instances. Section 4.4 is quite lengthy. Try to condense this section because it deviates

significantly from the original work. The same is true for Section 4.5. The authors must also

explain why the gel-based diet was superior to previous studies. This serves as the foundation

and highlights the work's uniqueness. The conclusion could be improved. Present what you have,

the gaps that have been identified, and how future research can be used to fill them.

Rest is fine, and I strongly urge the authors to carefully amend the manuscript based on the

suggestions above, as the work has enormous potential for furthering research in the field of AI

use in feed for improved reproduction and growth.

Reviewer #2: • Research topic is highly recommended for publishing with standard english writing. The whole manuscript is to be rechecked for grammatical errors.

• Growth and coloration of fishes are very much influenced by tank dimensions and light availability. Authors are advised to mention the type of tanks used during the experiments and the light arrangement in the experimental aquarium.

• Explanations may be incorporated for the selection of particular doses of letrozole (LET) during the current experiment.

• The nutritional composition of feed used during the current experiment needs to be incorporated to substantiate the unique proposal of gel feed for fishes.

• The following paragraphs need to be revised for grammar corrections and for better clarity about the topics

• 253- 254- How minimum mortality can be 82.22 % ?

• 229 to 234 – Grammatical corrections with clear descriptions are needed.

• 281-291 - Rewrite the sentence.

• 297-301- Grammatical corrections are required.

• The discussion part is poorly written and needs revision. The following sentences are lacking in meaning.

• 310-313

• 365-374

• 379-428

Conclusion to be amended for impressive illustrations of experiment.

6. PLOS authors have the option to publish the peer review history of their article (what does this mean?). If published, this will include your full peer review and any attached files.

Reviewer #1: **Yes: **Dr. Veerendra Singh

Reviewer #2: No

---

## [Author Response · Author response to Decision Letter 0]

3 Jun 2023

Respected Editor,

On behalf of the authors, I am very much thankful to the two anonymous reviewers for their insightful comments, constructive criticisms and suggestions which will enhance the quality of the submitted work. We have carefully read and tried our best to accommodate majority of the comments by the reviewers. As mentioned, the response to the specific comments is given below for kind consideration.

Response are attached as separate file.

---

## [Editor Report · Decision Letter 1]

15 Jun 2023

Effects of dietary aromatase inhibitors on Rosy Barb, Pethia conchonius, masculinization: evidence from growth, coloration, and gonado-physiological changes

PONE-D-23-08082R1

Dear Dr. Singh

We’re pleased to inform you that your manuscript has been judged scientifically suitable for publication and will be formally accepted for publication once it meets all outstanding technical requirements.

Kind regards,

Dharmendra Kumar Meena

Academic Editor

PLOS ONE

Additional Editor Comments (optional):

The article is recommended for acceptance for publication.
---

## [Editor Report · Acceptance letter]

29 Jun 2023

PONE-D-23-08082R1 

Effects of dietary aromatase inhibitors on masculinization of Rosy Barb (*Pethia conchonius*): evidence from growth, coloration and gonado-physiological changes 

Dear Dr. Singh:

I'm pleased to inform you that your manuscript has been deemed suitable for publication in PLOS ONE. Congratulations! Your manuscript is now with our production department. 

Kind regards, 

on behalf of

Dr. Dharmendra Kumar Meena 

Academic Editor

PLOS ONE